# Integrated near-field thermo-photovoltaics for heat recycling

Gaurang R. Bhatt [1]✉, Bo Zhao [2], Samantha Roberts[1], Ipshita Datta [1], Aseema Mohanty[1], Tong Lin[1], Jean-Michel Hartmann[3], Raphael St-Gelais[4], Shanhui Fan [2] & Michal Lipson [1]✉

Energy transferred via thermal radiation between two surfaces separated by nanometer distances can be much larger than the blackbody limit. However, realizing a scalable platform that utilizes this near-field energy exchange mechanism to generate electricity remains a challenge. Here, we present a fully integrated, reconfigurable and scalable platform operating in the near-field regime that performs controlled heat extraction and energy recycling. Our platform relies on an integrated nano-electromechanical system that enables precise positioning of a thermal emitter within nanometer distances from a room-temperature germanium photodetector to form a thermo-photovoltaic cell. We demonstrate over an order of magnitude enhancement of power generation ($P_{gen} \sim 1.25$ μWcm$^{-2}$) in our thermo-photovoltaic cell by actively tuning the gap between a hot-emitter ($T_E \sim 880$ K) and the cold photodetector ($T_D \sim 300$ K) from ~500 nm down to ~100 nm. Our nano-electromechanical system consumes negligible tuning power ($P_{gen}/P_{NEMS} \sim 10^4$) and relies on scalable silicon-based process technologies.

[1] Department of Electrical Engineering, Columbia University, New York, NY 10027, USA. [2] Department of Electrical Engineering, Ginzton Laboratory, Stanford University, Stanford, CA 94305, USA. [3] Université Grenoble Alpes, CEA, LETI, 38000 Grenoble, France. [4] Department of Mechanical Engineering, University of Ottawa, Ottawa, ON K1N 6N5, Canada. ✉email: grb2116@columbia.edu; ml3745@columbia.edu

Harvesting of thermal energy can be performed using radiation from systems with hot surfaces by placing a cold photodetector at nanometer distances. At nanometer (near-field) distances, strong heat exchange occurs due to evanescent modes, thermal radiation that cannot propagate from the hot body towards the far-field, but can evanescently couple from a hot to a cold surface when the separation is sub-wavelength[1–13]. Near-field radiative heat exchange have been demonstrated to overcome the blackbody limit at small gaps ($d$) between the hot and cold surfaces and scales as $1/d^{\alpha}$ ($1 \leq \alpha \leq 2$; $\alpha$ is a geometry-dependent factor). Potential applications of this effect include electricity generation on-demand from heat exchangers and thermal management systems in industrial and space applications[14–20].

Realizing a scalable platform that utilizes near-field heat transfer to generate electricity remains a challenge due to difficulty of fabricating two large-area surfaces separated by a small gap, while also simultaneously maintaining a large temperature differential between the surfaces as required for energy harvesting[21–29]. The gap between the surfaces needs to be small enough to induce near-field enhancement while preventing surface contact under the effects of intrinsic film stresses, thermal stresses, surface forces (Casimir forces, Van der Waals forces, etc.) and fabrication process variations[30,31]. While the underlying theory of heat exchange via near-field radiation has been widely studied over the past decade, it has only been recently demonstrated for energy generation using a lab-scale table-top experiment based on precision nano-positioning systems[32], or platforms relying on intermediate material spacers that limit the possible near-field enhancement[33].

## Results

We demonstrate a scalable near-field thermo-photovoltaic that allows us to control the heat flux through precise tuning of the distance between a hot suspended metallic thermal emitter and a room-temperature photodetector. The scalable platform is shown in Fig. 1a while the electron microscope image of an individual thermo-photovoltaic (TPV) cell is shown in Fig. 1b. The TPV cell consists of a suspended emitter, underlying photodetector and a pair of actuators for mechanical control. The suspended thermal emitter ($80 \times 15 \, \mu m^2$) consists of a metallic thin-film supported by a layer of amorphous silicon (a-Si) and anchored to silicon dioxide (SiO$_2$) pads with the help of multiple flexures (pad area: $A_{pad} \approx 450 \times 450 \, \mu m^2$, $t_{SiO_2} = 300 \, nm$). The suspended emitter overlaps the actuation electrode at each end ($A_{overlap} \approx 5 \times 15 \, \mu m^2$) while the rest of the bridge area overlaps the active region of a photodetector. The in-plane photodetector and the actuation electrode are separated by a distance of $4 \, \mu m$. The suspended emitter is brought closer to the underlying photodetector by virtue of electrostatic attraction through the application of an actuation potential $V_a$ to the actuators. The multiple flexures holding the central bridge of the emitter provide the required spring-like restoring force for controlled tuning of the gap. The details of the device dimensions are provided in Supplementary Note 1, and the fabrication procedure is provided in the "Methods" section.

We choose a tungsten-chrome thin-film thermal emitter and a germanium photodetector to demonstrate electricity generation from near-field radiative heat transfer. Fig. 2a-inset shows the cross-section of our TPV cell with various films stacks that constitute the emitter and the photodetector. We choose tungsten (W) as thermal emitter due to its thermal stability at high temperatures and compatibility with silicon processing technology. Thin-film chromium is required to achieve adhesion and nucleation of W films to the silicon bridge on the top and to the

underlying sacrificial layer during the fabrication process. Our choice of Ge photodetector is due to its relatively lower bandgap ($E_g \sim 0.67 \, eV$) that theoretically allows us over 16% spectral overlap (amounting to $\sim 62.5 \, mW \, cm^{-2}$) with our emitter at $T_E = 900 \, K$ as compared to a silicon ($E_g \sim 1.1 \, eV$). The Ge surface is covered with a thin-film of alumina ($t_{Al_2O_3} \approx 10 \, nm$) to avoid electrical shorting and damage to its active region. Plot of Fig. 2a shows the spectral heat flux from the metal emitter (Cr−W−Cr, $t_W \sim 80 \, nm$, $t_{Cr} \sim 5 \, nm$) as absorbed by a room-temperature Ge photodetector ($t_{Ge} \sim 2 \, \mu m$) placed 500 nm, 100 nm, and 50 nm apart computed using a fluctuational electrodynamics model (FED)[34–38]. The power absorbed by the photodetector increases significantly when the emitter and photodetector are separated by sub-wavelength distance. At larger gaps (far-field), the radiative power absorbed by the photodetector is much smaller due the limited thermal energy carried out by only the propagating modes. At sub-wavelength distances (near-field) the increase in absorbed power is due to evanescent coupling of thermal radiation from the hot-emitter to the cold photodetector. The details of the simulation model and the dispersion plots ($\omega - k$) for the far-field and near-field are provided in Supplementary Note 2. The computation results shown here are made at emitter temperature of 900 K and photodetector temperature of 293 K.

Our TPV design relies on thin-film metallic emitter with high-thermal resistance to achieve effective thermal insulation between the hot-emitter and the underlying photodetector, allowing heat transfer only through radiation. Figure 2b shows the heat map of the structure when the emitter is heated to high temperatures. One can see that while the emitter temperature is over 900 K, the temperature of the photodetector embedded in the substrate remains at 300 K. Maintaining such a large temperature difference between the emitter and the photodetector is essential for energy harvesting purposes. According to the Carnot efficiency limit $\eta_{Carnot} = (T_H - T_C)/T_H$ where, $T_H$ and $T_C$ are temperatures of hot and cold surfaces, respectively, the efficiency of a heat engine increases with a larger temperature difference between hot and cold sides. Furthermore, higher temperature differential also results in higher power density at the photodetector. The thermal circuit of the structure is provided in Supplementary Note 6. The heat map shown in Fig. 2b is computed using a finite element heat transfer model considering a full substrate thickness ($t_{sub} \sim 750 \, \mu m$) with its back surface held at constant temperature ($T_{back} \sim 293 \, K$). Effects of detector heating due to absorption of near-field heat exchange are appropriately taken into account (see Supplementary Note 6).

We show electrostatic control of gap, from 500 nm to 100 nm, between a suspended metallic emitter and the underlying Ge photodetector. We extract the displacement of the emitter surface by using the known initial gap ($d_0$) and measured change in capacitance caused by displacement of the bridge due to applied actuation potential ($V_a$). We confirm that the bridge is suspended and obtain the initial gap ($d_0 \sim 500 \, nm$) by performing an atomic force microscopy (AFM) measurement between the suspended bridge and the photodetector surface (see Supplementary Note 4). The measured initial gap is found to be more than the thickness of deposited sacrificial layer ($t_{SiO_2} \sim 300 \, nm$) due to bowing of the bridge under film-stress. The change in capacitance between the suspended bridge and the gate electrode as a function of actuation (emitter-gate) potential ($V_a$) is measured using a vector network analyzer[39–41]. The details of measurement setup are provided in the Supplementary Note 7. Using the initial capacitance ($C_0 = \epsilon_0 \cdot A_{actuator}/d_0$, where $\epsilon_0 = 8.85 \times 10^{-12} \, F \, m^{-1}$) and measured change in capacitance for different gate-emitter potential, the displacement of the suspended bridge-emitter is estimated. Figure 3 shows the gap between the emitter and the photodetector

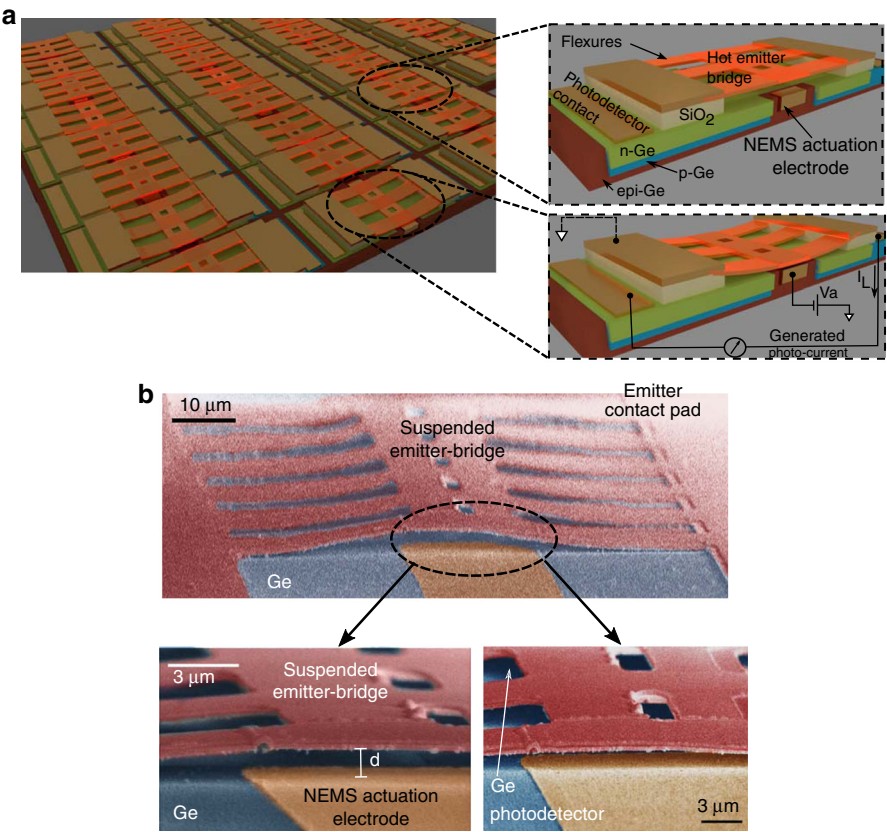

**Fig. 1 Near-field thermo-photovoltaic cell. a** Schematic of an array of thermo-photovoltaic cells demonstrated here. The platform consists of a nano-electromechanical system (NEMS) integrated with a photodetector cell. The NEMS consists of a suspended bridge ($80 \times 15$ μm²) made of amorphous silicon (a-Si) anchored on two silicon dioxide (SiO₂) pads. The zoomed in schematic shows a single unit cell, where one can see the suspended bridge, the partially underlying actuation electrode and the photodetector region. The silicon substrate over which the cell is fabricated is not shown here. The central region of suspended bridge is brought closer to the underlying photodetector cell through application of actuation potential. The suspended bridge serves as mechanical support for a thin-film metallic thermal emitter, which can be heated to high temperature. The generated electricity from near-field radiative heat is collected at the photodetector terminals. **b** Scanning electron micrograph of the fabricated structures showing the emitter suspended over the Ge photodetector and NEMS actuation electrode. The holes in the suspended bridge facilitate etching of the sacrificial oxide layer during the release process.

surfaces as a function of applied actuation potential $V_a$ ($0\,V \leq V_a \leq 1\,V$). One can see that the gap can be reduced from 500 nm down to ~180 nm for an applied potential of ~1 V. In order to ensure that observed change in capacitance is indeed due to displacement of suspended bridge-emitter, we perform another measurements using an unreleased device (sacrificial layer intact) and find negligible change in capacitance under influence of actuation potential. The measurement results are fitted with an electrostatic actuator model with non-linear spring response to estimate the displacement. The details of the fitting procedure along with the parameters and the raw data of the measured change in capacitance from a released and unreleased device are provided in Supplementary Note 7. The multiple flexures holding the central bridge, the film-stress and the partly overlapping actuation electrodes contribute to non-linear spring-like behavior allowing us to achieve gaps much smaller than those imposed by the pull-in limit of standard electrostatic actuator[42–44]. We initially restrict the calibration measurement of our nano-electromechanical system (NEMS) at 1 V to avoid any unwanted damage to the device prior to heat transfer measurements. The measurements shown here are performed under vacuum ($P_{chamber} < 8 \times 10^{-5}$ Torr) to overcome the limitation of low breakdown voltage of air ($V_b \sim 0.4$ V) at extremely small distances ($d \leq d_0$) in our devices.

We show an 11x (±2x) increase in generated electrical power as the distance between the suspended hot-emitter and the

underlying detector is reduced from ~500 nm to ~100 nm. We heat the thin-film emitter to high temperatures by passing electrical current ($P_{in} \approx 58$ mW). The temperature is estimated based on the independently measured coefficient of resistance ($\alpha \approx 1.6 \times 10^{-4}\,K^{-1}$) and real-time estimation of the initial resistance $R_0$. Details of temperature estimation are provided in Supplementary Note 7. During the process of heating the suspended emitter we ensure that the parasitic thermal conduction into the detector is negligible ($\Delta T_D \approx 17$ K), by estimating the change in detector temperature using its IV characteristics[45,46]. Calculations for estimating the detector temperature are provided in Supplementary Note 5. We apply a NEMS actuation potential ($V_a$) while maintaining the emitter temperature and simultaneously measure the IV characteristics of the detector. Figure 4a shows the measured IV characteristics of the Ge detector for $0 \leq V_a \leq 1.5$. As expected, the IV characteristics shift into the power generation quadrant (4th) with an increase in actuation potential as a result of reduction in emitter-detector distance and increase in tunneling of thermal photons[47]. In order to ensure that the measured change is indeed due to collected photons, we perform independent experiments and measure negligible contribution of electrical leakage paths such as actuator-to-detector, emitter-to-detector and NEMS leakage current. The raw-data for these measurements are provided in Supplementary Note 7 and 8. Moreover, the measurements shown here are limited to $V_a = 1.5$ V as beyond this bias-point our suspended structures undergo

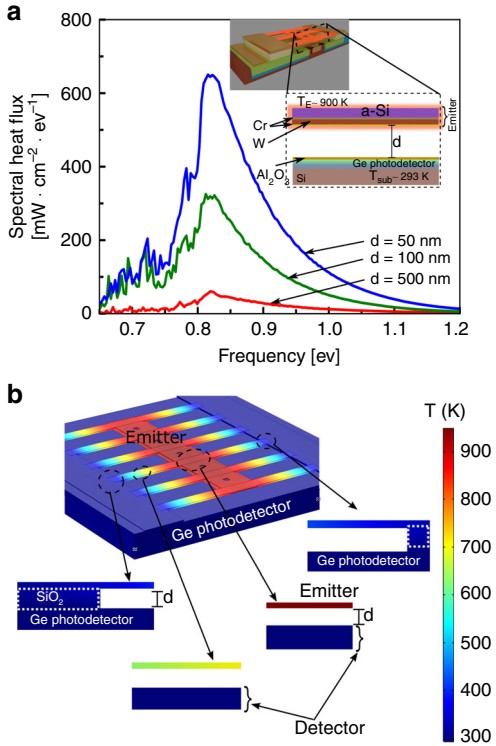

**Fig. 2 Computational analysis of near-field heat transfer in our thermo-photovoltaic cell. a** Calculated spectral heat flux of thermal radiation absorbed by room-temperature germanium when the hot-emitter (aSi-Cr-W-Cr) is placed at near-field distances (50 nm, blue-line, 100 nm, green line) and, far-away (500 nm, red-line). Inset shows: schematic of the material stack of emitter and photodetector of the thermo-photovoltaic cell. In the hot-stack, chromium (Cr) film helps in adhesion and nucleation of tungsten thin-film, while a-Si provides mechanical support when the bridge is suspended. The cold-stack is composed of Ge-on-silicon and covered with 10 nm alumina ($Al_2O_3$) to protect the surface and avoid any unwanted current leakage from bridge to photodetector in case of shorting during measurement. The temperature of the emitter is assumed to be 900 K. **b** Simulated temperature distribution in the structure when the emitter is heated up to high temperatures. The insets show the cross-section of the structure at different positions. One can see that the when the emitter is hot (shown by red) the clamped edges at the pad and the substrate remain at room-temperature (shown in blue), due to high-thermal resistance of the thin-film emitter.

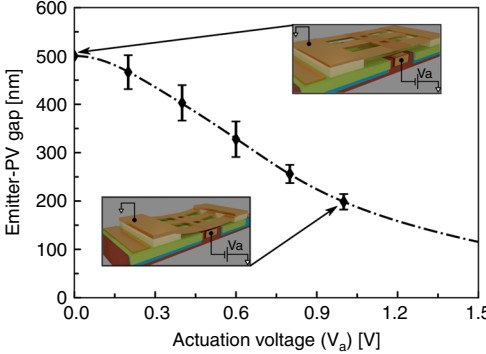

**Fig. 3 Nano-electromechanical switch characterization.** Displacement of suspended emitter as a function of applied gate-emitter potential difference ($V_a$). The broken line represents fit to a non-linear spring model (see Supplementary Note 7), while the scatter points show the experimental data.

unrecoverable damage due to excessive bending. The measurements shown in Fig. 4a are done under vacuum conditions (chamber pressure $< 8 \times 10^{-5}$) Torr) to minimize any convective heat losses. Inside the vacuum chamber, the chip rests on a large metallic chuck maintained at room-temperature. The schematic of the measurement setup is provided in "Methods" section.

We generate $\approx 1.25\ \mu W\ cm^{-2}$ from the near-field radiation between the hot Cr-W-Cr emitter at $T_E \sim 880\ K \pm 50\ K$ and the Ge photodetector at $T_D \sim 301\ K \pm 9\ K$. Figure 4b shows the generated power by our photodetector as a function of its distance from the suspended emitter. The collected power by Ge photodetector increases significantly (11x $\pm$ 2x) as the gap reduces from ~500 to ~100 nm. The generated power ($P_{gen}$) is given by the area of the largest rectangle that fits under each IV curve—$P_{gen} = FF \times V_{oc} \times I_{sc}$; $V_{oc}$ and $I_{sc}$ are the open-circuit voltage and short-circuit current estimated from the IV characteristics of Fig. 4a, respectively, while FF is the fill factor (FF ~ 0.25)[48]. Note that the power generated is much larger than the driving power for the capacitive NEMS structure shown here ($P_{gen}/P_{NEMS} = 10^4$; where $P_{NEMS} = 0.5 \cdot C_{NEMS} \cdot V_a^2$, $C_{NEMS}$ is the gate-emitter capacitance). The experimental data is seen to closely follow the theoretical prediction for the enhancement in heat transfer at small distances computed using FED. The computational data shown here takes into account the responsivity of our Ge photodetector and the bending of the emitter bridge under the influence of actuator bias $V_a$. The computation details are provided in Supplementary Note 2. The response of the NEMS at higher voltages ($V_a > 1\ V$) is extracted using the non-linear fit described earlier. The error-bars reported in generated power are computed based on multiple measurements of detector $I_{sc}$, while the error-bars reported in the gap-estimation are computed from the capacitance measured over a frequency band.

## Discussion

The overall efficiency of the TPV is governed primarily by the spectral mismatch between the emitter and the photodetector along with the photodetector quantum efficiency. In our experiments, the spectral overlap at emitter temperature of 880 K, is ~14% of the total emitted spectrum, equivalent to ~49 mW cm$^{-2}$ (see Supplementary Note 2). The overall spectral mismatch however, improves at higher emitter temperatures as more thermal radiation is concentrated above the detector bandgap, making it more suitable for high-temperature energy harvesting. Moreover, surface roughness of the emitter and detector is also known to modify the spectral overlap[49,50]. In our case the measured root-mean-squared roughness of the emitter and detector surface is found to be $\sigma_{Emit} \sim 1.7\ nm$ and, $\sigma_{Det} \sim 1.1\ nm$, respectively. Our theoretical calculations suggest that the spectral overlap takes an extra ~8% penalty due to the surface roughness (see Supplementary Note 3). The thickness of detector-layer (Ge) also plays an important role in total power absorbed with ~12% reduction for a 10% reduction in thickness. The performance of our TPV cells is also limited by the low efficiency ($\eta_D \approx 0.3 \times 10^{-4}$ %) of Ge photodetector arising from high recombination in the doped Ge regions (see Supplementary Note 9) possibly arising from dislocation defects at the epitaxial growth interface[51].

Our reconfigurable TPV platform, demonstrated using standard silicon fabrication process technology, can be scaled for harnessing waste heat by proper engineering of heat channels from source to emitter[52]. Considering the current state-of-art germanium photovoltaic detectors with quantum efficiency of over 10%, we can significantly improve the overall efficiency of our TPV (>4 mW cm$^{-2}$)[53]. Leveraging the NEMS technology also allows us to overcome fabrication limitations by making TPV cells with larger initial gaps between emitter and detector, which

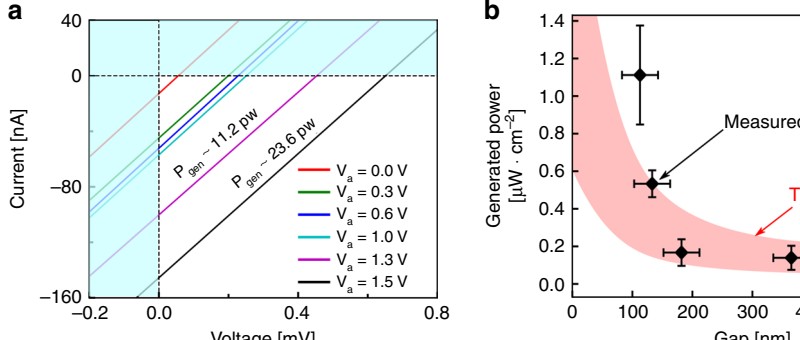

**Fig. 4 Characterization of thermo-photovoltaic cell. a** Measured change in IV for constant emitter temperature and varying gap achieved by applying NEMS actuation potential ($V_a$). The IV curves shift into 4th quadrant representing power generation from the near-field thermal radiation. Farther the shift, more is the power generated. **b** Measured power density from our thermo-photovoltaic (TPV) cell as a function of emitter-detector gap. The error-bars on x-axis represent the mean error in estimation of displacement while, error-bars on y-axis represent the variation in power measured over two successive measurements on the same device. Also shown is a confidence interval of the computed power generation for our TPV cell with independently measured efficiency of our Ge photodetector accounted. The simulation curves also account for the bending of the suspended structure under the influence of actuation potential that is applied to reduce its distance with respect to the underlying photodetector. The simultaneously measured emitter-to-actuator leakage current is found to be <16 pA—nearly four orders smaller than the current measured from the thermal radiation.

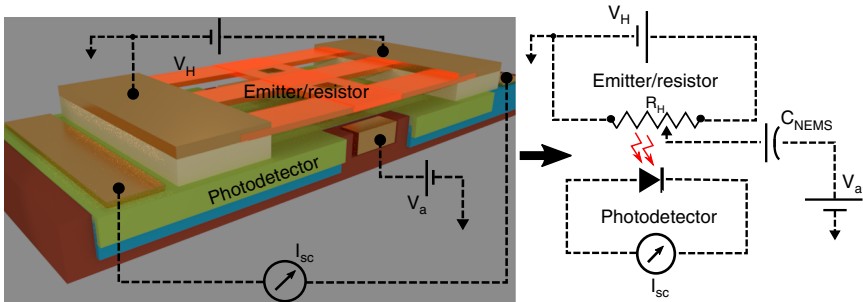

**Fig. 5 Heat transfer measurement setup.** External circuit connections for the heat transfer measurements shown in Fig. 4 along with the equivalent electrical circuit of the device and the measurement setup. The emitter is applied a potential $V_H$ to heat up, and the detector current ($I_{sc}$) is measured for different NEMS actuation potentials ($V_a$). In the equivalent circuit, the NEMS actuator is represented by capacitor $C_{NEMS}$, the emitter as electrical resistor $R_H$ and, the photodetector as diode.

can then be brought closer electrostatically with minimal operating power. A reconfigurable NEMS platform will also offer on-demand heat recycling and temperature control for systems with constant temperature fluctuations[54]. Reconfigurable NEMS-based TPV platforms can find applications in dynamic heat recovery and cooling systems employed in data centers, as well as metal foundries[55].

## Methods

**Fabrication process.** Ge photodetector cell fabrication: germanium photodetector is fabricated on a silicon handle wafer. The epitaxial Germanium (Ge) layer ($t_{Ge}$ ~ 1.8 μm) is grown using $GeH_4$ on a standard 8″ silicon wafer in a reduced pressure-chemical vapor deposition chamber[56]. A low-temperature/high-temperature strategy is used, together with some short duration thermal cycling, to obtain a rather flat, slightly tensile strained and high crystalline quality Ge layer, with a threading dislocations density around $1 \times 10^7$ cm$^{-2}$. After this the patterns for defining p-doping regions are transferred and then subsequently doped with boron (species: $11B^+$, dose: $5 \times 10^{13}$ cm$^{-2}$, power: 33 KeV). Afterwards, the p-doped regions are patterned again to define n-well regions, which are then implanted with phosphorous (species: 31 $P^+$, dose: $1 \times 10^{15}$ cm$^{-2}$, power: 50 KeV). During the implantation process the epi-Ge layer outside the patterned area is protected using ~60 nm silicon dioxide ($SiO_2$) layer deposited using plasma enhanced chemical vapor deposition (PECVD). The resulting p–n junction diode is formed in the direction perpendicular to the wafer surface. Metal contacts for the p–n diode are then deposited using e-beam evaporation technique for Ni as contact metal and Pt as the cover metal ($t_{Ni}$ ~ 20 nm, $t_{Pt}$ ~ 45 nm). Rapid thermal annealing is then performed to fix lattice defects from implants, as well as to allow metal contact formation. The TPV cells with integrated nano-electromechanical system are fabricated on top of the Ge photodetector.

NEMS fabrication: in-plane actuation electrode region is carefully patterned, etched and filled up with platinum and no electrical short-circuit with in-plane diode ensured. The detector and gate are covered with sacrificial layer of thin-film $SiO_2$ ($t_{SiO_2}$ ~ 300 nm) using PECVD. On top of this sacrificial layer emitter patterns with their contact pads are created and tungsten (W) films ($t_W$ ~ 80 nm) are sputtered (pressure: 12 mT, power: 290 W, Ar: 30 sccm) and a lift-off procedure is performed. In order to overcome the issue of adhesion and nucleation of tungsten films, we deposit thin-film of chromium ($t_{Cr}$ ~ 5 nm) before and after the deposition of W. On top of patterned Cr-W-Cr emitters, we deposit amorphous silicon (a-Si) using PECVD (table temp: 350 ˚C, $SiH_4$ (1) +Ar (9) = 210 sccm, pressure: 1000 mT, RF power: 10 W) to provide mechanical stiffness when the bridge is suspended. The a-Si films are patterned and etched to match the underlying W emitters. The bridges are then suspended by a release step where the sacrificial $SiO_2$ is under-etched using vapor HF (SPTS Primaxx uEtch). The thin Cr layer acts as a protective layer for W-films while $Al_2O_3$ helps protect the Ge surface during the release step.

**Measurement procedure.** The measurements are performed inside a vacuum chamber using dedicated source meters (Keithley 2400) for applying heating power to the emitters, measurement of detector characteristics (IVs), and applying NEMS gate bias. The circuit connections are shown below in Fig. 5. The heating power is gradually applied to the heaters while simultaneously measuring the change in resistance of the heaters, change in diode temperature and any electrical leakage between emitter and NEMS gate. Once the desired temperature is achieved (i.e., temperature at which detector current is high enough to continue the NEMS measurements), the NEMS gate potential is applied and diode IV characteristics are recorded. The source meter used for measuring the diode IV characteristics is set to averaging 25 data samples with an integration time of ~80 ms per sample. The Fig. 5(inset) shows the equivalent circuit of the TPV device along with external source meter connections.

## Data availability

The data that support the findings of this study are available within the article and the Supplementary Information file.

## Code availability

Simulation codes are open-access available, details can be found in the Supplementary Information file reference list. Details of data-analysis and fitting methods are provided in the Supplementary Information file.

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

## Acknowledgements

M.L., S.F., G.B., and B.Z. acknowledge the financial support from ARPA-E IDEAS program (#DE-AR0000731). R.S. acknowledges the financial support from the Natural Sciences and Engineering Research Council of Canada (NSERC). We also acknowledge the use of facilities at Advanced Science Research Center (ASRC) at City University of New York (CUNY), Columbia Nano-initiative, Cornell Nano-Scale Facility (NSF, #

NNCI-1542081) and Cornell Center for Materials Research (NSF MRSEC, # DMR-1719875). We acknowledge Prof. J. Hone at Columbia University for allowing the use of vacuum probe station measurements facility at his labs. We thank Y. Chang at National Chiao Tung University, A. Dutt at Stanford University and S.A. Miller, M. Zadka, B. Lee, U.D. Dave, C.S. Joshi, J.R. Rodrigues at Columbia University, for useful discussions. We also thank V. Narang at ASRC–CUNY for their help during AFM measurements.

## Author contributions

R.S., G.B., S.F., and M.L. conceived the design of the TPV. G.B. and R.S. designed the TPV cell. B.Z. and G.B. performed the heat transfer simulations and theoretical analysis. G.B. performed the NEMS simulations, device design, and fabrication flow and optimization. J.H. provided the epitaxial germanium on silicon. G.B. and S.R. performed the fabrication of TPV cells. G.B. performed the TPV measurements, device characterizations, and data analysis. G.B. and I.D. performed the VNA measurements for NEMS characterization. I.D., T.L., and A.M. provided critical feedback at different stages of device design, experiments and data analysis. G.B. and M.L. prepared the manuscript. S.F., B.Z., R.S., I.D., A.M., S.R. edited and provided feedback on the manuscript.

## Competing interests

The authors declare no competing interests.
