## [Peer Review File · Nature Communications]

Reviewers' comments:

Reviewer #1 (Remarks to the Author):

This manuscript reports measurements of heat-to-electricity conversion via integrated near-field TPVs, where the high-temperature emitter and low-temperature photodetector are separated by vacuum gaps from ~ 500 nm down to ~ 100 nm. The contribution of evanescent modes is quantified by comparing experimental data against theoretical predictions based on fluctuational electrodynamics. It is shown that the power generated is enhanced due to evanescent wave coupling at sub-wavelength vacuum gap distances.

The topic of this paper is interesting and timely. While a number of papers have experimentally demonstrated the enhancement of the radiative flux beyond the blackbody limit in the near field, experimental demonstration of TPV power generation capitalizing on thermally generated evanescent modes is very limited. The approach used in the paper (i.e., integrated near-field TPV platform) is novel, and different from the current state-of-the-art. Fig. 1a suggests that the proposed platform is scalable to real-world application, which is in contrast with current laboratory-scale near-field TPV experiments. I therefore support publication of this manuscript in Nature Communications after the authors address the following questions/comments:

1. Through simulations, Fig. 1c shows that the photodetector remains at low temperature (~ 300 K) even if the emitter is at high temperature (~ 900 K). I am confused by the temperature of the photodetector, both in the simulations and experiments:

a) In the simulations, what are the boundary conditions? Are you assuming that the back of the photodetector is at constant temperature of ~ 300 K? If not, are you using a convective boundary condition?

b) In the experiments, are you actively cooling the photodetector? If not, can you describe how the near-field TPV platform is heat sinked?

2. What is the surface roughness of the Cr and Al₂O₃ layers? Is the roughness negligible for the theoretical calculations?

3. Fig. 2 (and Fig. S2 in supplementary section 2) shows near-field radiative heat transfer simulations for gap distances of 50 nm and 50 μ m. For consistency with the experimental results, I suggest to replace the 50 nm results by theoretical predictions at a gap distance of 100 nm.

4. The absorption bandgap of the Ge photodetector (~ 0.67 eV) is, as mentioned in the paper, lower than conventional Si cells (~ 1.1 eV). Yet, for an emitter temperature of ~ 900 K, only a small portion of the thermal spectrum is above the Ge photodetector bandgap. Can you comment on this? How the spectral mismatch between the emitter and photodetector affects near-field TPV performance?

5. In Fig. 4b, why do you have theoretical data for 2 values of TH? My understanding is that TH_{average} is ~ 880 K, with an uncertainty of ± 50 K. Then, why not plotting a colored band of theoretical prediction instead of 2 distinct curves?

6. Is the efficiency provided in line 192 in %? Do you think that the low efficiency of near-field TPVs can be explained (partially at least) by the spectral mismatch between the emitter and photodetector?

Minor comments:

7. Lines 28-30. The first sentence of the paper is not clear. For example, what do you mean by "dynamic temperature profiles"?

8. Line 31. Evanescent modes are emitted, regardless of the regime at which radiative transfer takes place (i.e., far field versus near field). As such, I would change "be emitted" by "propagate".

9. Line 39. What do you mean by "non-configurable passive platforms"?

10. Line 104. Why do you call propagating waves "propagating guided modes"? More specifically, why do you use the word "guided"?

11. It is mentioned twice in the manuscript (lines 78 and 150) that the equivalent thermal circuit is provided in supplementary section 5. Only one reference to supplementary section 5 is sufficient.

12. Line 54 of the supplementary material. Change "estimate" by "estimated".

Reviewer #2 (Remarks to the Author):

The work described in the manuscript by Bhatt et al. (entitled "Integrated near-field thermo-photovoltaics for on-demand heat recycling") is, broadly speaking, directed towards the development of near-field thermo-photovoltaic energy conversion devices. Recent theoretical and experimental work (mostly) by others in the community has demonstrated the benefits of leveraging near-field effects in this energy conversion process to dramatically increase the heat-to-electricity conversion efficiency (some of the relevant papers are cited by the authors)

The authors assert that "The challenge lies in designing a platform that can separate two surfaces by a small and tunable gap while maintaining a large temperature differential." They furthermore imply that a reconfigurable platform for on-demand energy conversion is essential and finally claim that they have developed a "fully integrated, reconfigurable and scalable platform" that is suitable for nearfield-based thermoelectric energy conversion.

While nearfield thermophotovoltaics (TPV) has (re-)emerged as an interesting concept, the central challenge in the field is to develop the fundamental understanding and implement systems (nearfield TPV cells) that enable heat-to-electricity conversion that is at least comparable in efficiency to existing (commercial) thermoelectric devices. The proposal by the authors that reconfigurable platforms and on-demand heat recycling are central to TPV is completely misguided and wrong. Using NEMS actuator to readjust the distance between the emitter and photodetector is far too complex and expensive, and it is of no interest for controlling the power output from a TPV platform (as implied by on-demand in the title). To "turn the system off" one should (simply) remove the flow of heat to the TPV device (the only real option to avoid any additional inefficiency). Additionally, the generated electricity is ideally suited for switching on and off applications.

In the study of nearfield radiation and TPV (that means in fundamental science and engineering studies) continuous adjustments of the distance between the emitter and photodetector have (and will continue) to play a central role in the detailed analysis of the processes under investigation. However, the platform described by Bhatt et al. falls far short of providing a suitable and precise measured range of distances to critically evaluate device function. In fact, the distance data reported (Fig 3 and 4) lack the independent confirmation and accuracy of previous studies (see for example ref. 32). Is the work novel and of interest to the community?

The work is original in that it applies NEMS actuation to the design of a TPV system, but, as I outlined above, this is an unnecessary and misguided use that will not benefit the development of efficient, successful TPV systems. In addition, electrostatic NEMS actuation as described in this paper is not novel and has been leveraged in the senior author's lab in the study of nearfield radiation previously (ref. 24). The reported platform is, even by the standards of this young field, inefficient ($\sim 0.3 \cdot 10^{-4}$) as compared to recently published work (for example ref. 32), and the assertion by that authors that the electrostatic control of the NEMS actuator consumes little power is not reassuring, because the actuation is not desirable or improving practical TPV platforms.

With regard to the writing and presentation quality of the manuscript, it is difficult to fully comprehend the design of the NEMS system in relation to the photodetector and critically evaluated important aspects of the platform, including the thermal design and nanometer-level actuation of the system. Critical data, for example, on the cleanliness (particle contamination) and flatness of the devices are missing, and I am not at all convinced that the device is truly scalable to large areas. Figure 1 is not clear, there should be a schematic drawing. In addition, the background information (relative to nearfield radiation, TPV and electrostatic NEMS actuation) lacks balance in the presentation of past work by other groups, and the manuscript falls short in critically discussing the accomplishments made in this work. Also, the use of the term reconfigurable appears inappropriate in the context of what past work on reconfigurable engineering systems is referring to. In this manuscript the authors demonstrated limited adjustability of a single parameter – the emitter to photodetector distance. In summary, the paper offers no meaningful progress in near-field TPV. Consequently, it will not impact the thinking in the field and I cannot support publication of the manuscript.

Reviewer #3 (Remarks to the Author):

The topic of the manuscript, namely that of heat recycling using controlled near-field thermophotovoltaics, is very important and relevant, enabled by the recent advances in precision nanofabrication technologies. Indeed, the nanostructure complexity and precision in the control of the distance between the heater and the photodetector at the nanoscale, demonstrated in this manuscript, are impressive. The main results, reported in Fig 4b, show a superlinear increase in the generated power versus the decreasing gap distance. The measurement is carried out in 6 points only with rather large error bars both in the abscissa and ordinate axes. I apologize for my possible omission, but I couldn't see how were the error bars considered in the report of the measurement? In particular, why does the error bar grow with the increase in the signal amplitude (generated power)? Furthermore, two graphs based on the calculation (dotted red and blue lines) are reported to give the 'confidence interval' on the temperature of the hot side. However, I see that neither of the calculated curves fits the entire data set in a satisfactory manner (even including large error bars). Does this indicate systematic issues with the measurement of the generated power or the estimate of the hot temperature of the metal or both? I believe authors should offer some explanations for the large discrepancy between the measurement and the theory, as it relates directly to the main presented dataset.

The interpretation of the data strongly relies on the knowledge of the hot (T_{hot}) and cold (T_{cold}) reservoir temperatures. To the best of my knowledge, I did not see a reported measurement of the T_{hot} in the manuscript nor supplemental information (only an estimate based on the dissipated power into the W strip). The cold side is estimated in temperature from model fits of the I-V curves. I do not know what instrumentation is available to the authors, but given how important is the knowledge of these temperatures, I would like to see a more elaborated discussion of the measurement(s)/estimate(s)/fit(s) of these quantities, together with the confidence intervals (especially, in connection to the discussion of Fig 4b above). Given large error bars in the data, I

believe that the claimed factor of 11 enhancement in the generated electrical power (in the near-field), should be specified with the experimental certainty bounds as well.

On a similar note, I found it quite astonishing that the temperature difference between a ~ 900K and 300K body can be maintained so well when the bodies are in a perfect mechanical contact with each other over few-micron distances. As authors point out, this should only be possible by a "thin-film metallic emitter with high thermal resistance" and I think this is a discovery in itself that the authors found such a highly efficient thermal insulation. For clarity, I would like to understand about the details of the measurements. Are these steady-state measurements or obtained via modulation techniques (e.g. lock-in amplification)? Also, on what timescales are these measurements taken?

I would like to request mandatory revisions addressing the points raised. I believe the manuscript is otherwise very well written, clear and addresses an interesting topic, worthy of the attention of the broad readership of Nature Communications.

We thank all the reviewers for their insightful, constructive and, critical comments that have helped us improve our manuscript. All the questions are addressed in blue-faced-fonts. The major changes and additions in the manuscript are listed below. The changes have also been highlighted in the manuscript.

List of Major Changes:

- 1) Main-text: Re-worked error bars based with raw data provided in supplementary
- 2) Main-text: Discussion after experimental results summarizing all queries
- 3) Methods: Explanation of experimental setup with schematic
- 4) Supplementary: Surface roughness measurements and its contribution to heat transfer
- 5) Supplementary: Measurement data and fit details on the estimation of emitter temperature
- 6) Supplementary: Gap estimation – measurement data

Reviewers' comments:

Reviewer #1 (Remarks to the Author):

This manuscript reports measurements of heat-to-electricity conversion via integrated near-field TPVs, where the high-temperature emitter and low-temperature photodetector are separated by vacuum gaps from ~ 500 nm down to ~ 100 nm. The contribution of evanescent modes is quantified by comparing experimental data against theoretical predictions based on fluctuational electrodynamics. It is shown that the power generated is enhanced due to evanescent wave coupling at sub-wavelength vacuum gap distances.

The topic of this paper is interesting and timely. While a number of papers have experimentally demonstrated the enhancement of the radiative flux beyond the blackbody limit in the near field, experimental demonstration of TPV power generation capitalizing on thermally generated evanescent modes is very limited. The approach used in the paper (i.e., integrated near-field TPV platform) is novel, and different from the current state-of-the-art. Fig. 1a suggests that the proposed platform is scalable to real-world application, which is in contrast with current laboratory-scale near-field TPV experiments. I therefore support publication of this manuscript in Nature Communications after the authors address the following questions/comments:

Response:

We thank the reviewer for their insightful comments. We have addressed all the queries in a point-by-point basis in our response below.

1. Through simulations, Fig. 1c shows that the photodetector remains at low temperature (~ 300 K) even if the emitter is at high temperature (~ 900 K). I am confused by the temperature of the photodetector, both in the simulations and experiments:

- a. In the simulations, what are the boundary conditions? Are you assuming that the back of the photodetector is at constant temperature of ~ 300 K? If not, are you using a convective boundary condition?

In our simulations on temperature distribution (Fig. 1d), we have considered the backside of the silicon handle wafer (thickness ~ 500 μm), housing the photodetector and the emitter, at a constant temperature of 293 K. We have not considered any convection for the surrounding of the device. The gap between the suspended emitter and the underlying detector is modeled as a material with thermal conductivity $k_{gap} \sim d \cdot q_{NF} / (A \cdot \Delta T)$. Where, d is the emitter-detector gap (~ 100 nm), A is the emitter-detector overlap area, q_{NF} is the heat flux above detector bandgap when emitter is at 900 K and, ΔT is the temperature difference.

The information is now added in the main text (Pg. 3, last sentence above Fig. 2), and supplementary section 6.

- b. In the experiments, are you actively cooling the photodetector?

In the experiment, we do not actively cool the chip. The temperature of the photodetector is ascertained by measuring its IV characteristics, when the emitter is cold and hot. The measurement details with non-ideal diode model fit are provided in supplementary section 5.

- c. If not, can you describe how the near-field TPV platform is heat sink-ed?

During the measurements chip is lying on a large temperature monitored cylindrical metal chuck (dia ~ 3 inch, height ~ 1.5 inches) at room temperature inside a vacuum chamber, which provides sufficient thermal capacity acting as a heat sink for the chip. The measurement is performed under vacuum ($P < 8 \times 10^{-5}$ Torr) to ensure there is no other convective cooling effects on the heaters.

The information is added in the main text (Pg. 6, last sentence above Fig. 4).

2. What is the surface roughness of the Cr and Al₂O₃ layers? Is the roughness negligible for the theoretical calculations?

The roughness of the Cr layer is $\sigma_{Emit} \sim 1.7$ nm and that of the Al₂O₃ layer is $\sigma_{Det} \sim 1.1$ nm. Our theoretical calculations suggest that the roughness induces $\sim 8\%$ reduction in overall heat-flux above detector bandgap as compared to a smooth surface ($\sigma_{Emit} = \sigma_{Det} = 0$) when the gap is 100 nm.

The calculations are performed by splitting a rough surface into multilayer medium and subsequently using Maxwell-Garnet mixing formulae to estimate its effective permittivity (Markel, 2016; Xu et al., 2019). We use the effective permittivity in the vicinity of the surface while still maintaining the gap at 100 nm. The heat transfer is computed as described in supplementary section 1, using the multilayer model based on fluctuational electrodynamics. The simulated heat-flux using the measured roughness of detector and emitter surfaces is shown here in Fig. 1 together with a smooth surface case ($\sigma_{Emit} = \sigma_{Det} = 0$). We note that the effect of surface roughness decreases at larger gaps.

Figure 1. Computed spectral heat flux for our TPV at 100 nm gap between the emitter and the detector for smooth (solid line, $\sigma_{Emit} = \sigma_{Det} = 0$) and rough case ($\sigma_{Emit} \sim 1.7$ nm, $\sigma_{Det} \sim 1.1$ nm, broken line).

The AFM data and theoretical simulations showing contribution of roughness to heat-transfer are added in supplementary section 3. The references related to this calculation are added in the supplementary reference list.

- Fig. 2 (and Fig. S2 in supplementary section 2) shows near-field radiative heat transfer simulations for gap distances of 50 nm and 50 μ m. For consistency with the experimental results, I suggest to replace the 50 nm results by theoretical predictions at a gap distance of 100 nm.

We have revised the Fig 2b and supplementary section 2 and replaced the theoretical plot showing heat-transfer for $d=50$ nm by $d=100$ nm.

- The absorption bandgap of the Ge photodetector (~ 0.67 eV) is, as mentioned in the paper, lower than conventional Si cells (~ 1.1 eV). Yet, for an emitter temperature of ~ 900 K, only a small portion of the thermal spectrum is above the Ge photodetector bandgap. Can you comment on this? How the spectral mismatch between the emitter and photodetector affects near-field TPV performance?

We thank the reviewer for their insightful comment. We indeed agree that the spectral mismatch between the emitter and the photodetector plays an important role in the performance of a TPV device along with the photodetector quantum efficiency. In our case, the spectral overlap at 880 K, is ~ 14 % of the total emitted spectrum, equivalent to ~ 49 $\text{mW} \cdot \text{cm}^{-2}$. The spectral matching improves at higher emitter temperatures as more energy from the emitter is concentrated above the detector bandgap. The thickness of the detector-layer (Ge) also plays an important role in total spectral absorption. Our computations suggest that $\sim 10\%$ reduction in Ge thickness reduces the power absorbed by $>12\%$.

The spectral overlap is computed by taking a ratio of power absorbed by our germanium detector over total power absorbed by all layers on the detector side. The plot of this calculated spectral efficiency as a function emitter temperature is shown below. The choice of our emitter and detector makes our platform suitable for high-temperature heat-recovery systems.

Figure 2. (a) Spectral overlap and, (b) power absorbed in Germanium detector as a function of emitter temperature.

5. In Fig. 4b, why do you have theoretical data for 2 values of TH? My understanding is that TH, average is ~ 880 K, with an uncertainty of ± 50 K. Then, why not plotting a colored band of theoretical prediction instead of 2 distinct curves?

We acknowledge the reviewer's comments and have revised the plot in Fig. 4b. The theoretical data is replaced by a colored band showing confidence bound interval $\sim 880 \pm 50$ K.

6. Is the efficiency provided in line 192 in %? Do you think that the low efficiency of near-field TPVs can be explained (partially at least) by the spectral mismatch between the emitter and photodetector?

We thank the reviewer for pointing this typo. The efficiency in line 192 is indeed in %. The mistake is now corrected.

We agree that the spectral mismatch between the emitter and the photodetector plays an important role in the performance of a TPV device along with the photodetector quantum efficiency. In our case, at ~ 880 K of emitter temperature, the spectral overlap between the emission spectrum of emitter and absorption of the detector is ~ 14 %, equivalent to $\sim 49 \text{ mW} \cdot \text{cm}^{-2}$.

In our case, our quantum efficiency ($\eta_D \approx 0.3 \times 10^{-4}$ %) is limited by the detector's low open-circuit voltage (V_{OC}), arising from high recombination rate in the Ge substrate possibly arising from dislocation defects at the epitaxial growth interface (Reboud et al., 2017). Considering the current state-of-art Ge PV quantum efficiency of over 10%, we can significantly improve ($> 4 \text{ mW} \cdot \text{cm}^{-2}$) the overall efficiency of our TPV (Korun and Navruz, 2016).

Minor comments:

7. Lines 28-30. The first sentence of the paper is not clear. For example, what do you mean by "dynamic temperature profiles"?

Thermal management systems in data-centers as well as metallurgical foundries often encounter dynamically changing hotspots or temperature fluctuations that require immediate cooling to avoid unwanted damage (Boucher et al., 2006). Near-field heat transfer based heat recovery systems could be significantly more efficient for cooling as the heat-flux could be higher than that observed in conventional conduction based systems (DeSutter et al., 2019; Guha et al., 2012; Kloppstech et al., 2017).

We have added the explanation in the discussion after reporting main-results in the manuscript. The term is removed from the introduction section to avoid confusion.

8. Line 31. Evanescent modes are emitted, regardless of the regime at which radiative transfer takes place (i.e., far field versus near field). As such, I would change "be emitted" by "propagate".

We have replaced emitted by propagate.

9. Line 39. What do you mean by "non-configurable passive platforms"?

We thank the reviewers' for pointing out the confusion in the manuscript. In this case, non-reconfigurable platforms are those that don't allow adjusting the heat-transfer rate through modification of distance between the emitter and the detector. As described earlier in Q7, such platforms can be used for heat-recovery systems employed for cooling.

This point is now clarified in the main-text in the discussion paragraph added on page 7 of the manuscript.

10. Line 104. Why do you call propagating waves "propagating guided modes"? More specifically, why do you use the word "guided"?

We have corrected this to propagating modes in accordance to point 8 above.

11. It is mentioned twice in the manuscript (lines 78 and 150) that the equivalent thermal circuit is provided in supplementary section 5. Only one reference to supplementary section 5 is sufficient.

We have removed the repetition.

12. Line 54 of the supplementary material. Change "estimate" by "estimated".

We have corrected this grammatical error.

Reviewer #2 (Remarks to the Author):

The work described in the manuscript by Bhatt et al. (entitled "Integrated near-field thermo-photovoltaics for on-demand heat recycling") is, broadly speaking, directed towards the development of near-field thermo-photovoltaic energy conversion devices. Recent theoretical

and experimental work (mostly) by others in the community has demonstrated the benefits of leveraging near-field effects in this energy conversion process to dramatically increase the heat-to-electricity conversion efficiency (some of the relevant papers are cited by the authors). The authors assert that “The challenge lies in designing a platform that can separate two surfaces by a small and tunable gap while maintaining a large temperature differential.” They furthermore imply that a reconfigurable platform for on-demand energy conversion is essential and finally claim that they have developed a “fully integrated, reconfigurable and scalable platform” that is suitable for nearfield-based thermoelectric energy conversion. While nearfield thermophotovoltaics (TPV) has (re-)emerged as an interesting concept, the central challenge in the field is to develop the fundamental understanding and implement systems (nearfield TPV cells) that enable heat-to-electricity conversion that is at least comparable in efficiency to existing (commercial) thermoelectric devices. The proposal by the authors that reconfigurable platforms and on-demand heat recycling are central to TPV is completely misguided and wrong. Using NEMS actuator to readjust the distance between the emitter and photodetector is far too complex and expensive, and it is of no interest for controlling the power output from a TPV platform (as implied by on-demand in the title). To “turn the system off” one should (simply) remove the flow of heat to the TPV device (the only real option to avoid any additional inefficiency). Additionally, the generated electricity is ideally suited for switching on and off applications.

In the study of nearfield radiation and TPV (that means in fundamental science and engineering studies) continuous adjustments of the distance between the emitter and photodetector have (and will continue) to play a central role in the detailed analysis of the processes under investigation. However, the platform described by Bhatt et al. falls far short of providing a suitable and precise measured range of distances to critically evaluate device function. In fact, the distance data reported (Fig 3 and 4) lack the independent confirmation and accuracy of previous studies (see for example ref. 32).

Is the work novel and of interest to the community?

The work is original in that it applies NEMS actuation to the design of a TPV system, but, as I outlined above, this is an unnecessary and misguided use that will not benefit the development of efficient, successful TPV systems. In addition, electrostatic NEMS actuation as described in this paper is not novel and has been leveraged in the senior author’s lab in the study of nearfield radiation previously (ref. 24). The reported platform is, even by the standards of this young field, inefficient ($\sim 0.3 \cdot 10^{-4}$) as compared to recently published work (for example ref. 32), and the assertion by that authors that the electrostatic control of the NEMS actuator consumes little power is not reassuring, because the actuation is not desirable or improving practical TPV platforms.

With regard to the writing and presentation quality of the manuscript, it is difficult to fully comprehend the design of the NEMS system in relation to the photodetector and critically evaluated important aspects of the platform, including the thermal design and nanometer-level actuation of the system. Critical data, for example, on the cleanliness (particle contamination)

and flatness of the devices are missing, and I am not at all convinced that the device is truly scalable to large areas. Figure 1 is not clear, there should be a schematic drawing. In addition, the background information (relative to nearfield radiation, TPV and electrostatic NEMS actuation) lacks balance in the presentation of past work by other groups, and the manuscript falls short in critically discussing the accomplishments made in this work. Also, the use of the term reconfigurable appears inappropriate in the context of what past work on reconfigurable engineering systems is referring to. In this manuscript the authors demonstrated limited adjustability of a single parameter – the emitter to photodetector distance. In summary, the paper offers no meaningful progress in near-field TPV. Consequently, it will not impact the thinking in the field and I cannot support publication of the manuscript.

Response:

We thank the reviewer for their critical comments. The concerns are addressed in a point-by-point basis below.

“... Using NEMS actuator to readjust the distance between the emitter and photodetector is far too complex and expensive, and it is of no interest for controlling the power output from a TPV platform (as implied by on-demand in the title).”“The work is original in that it applies NEMS actuation to the design of a TPV system, but, as I outlined above, this is an unnecessary and misguided use that will not benefit the development of efficient, successful TPV systems. ...”

An effective TPV system requires thin-film emitters (with low thermal capacity) separated by sub-100 nm gaps from detectors spanning over tens of microns lengths (large heat radiation area). Emitters with such large aspect-ratios (*thickness* \ll *length,width*), held at fixed distance, are difficult to realize due to variations in intrinsic film stresses, thermal stresses, surface forces (Casimir forces, Van der Waals forces, etc.) and, fabrication processes that results in nanoscale gaps to significantly vary (Serry et al., 1998). Thin-film stresses, often higher than the bulk material itself, lead to buckling with deformations (blister height) exceeding 100 nm, therefore collapsing the emitter-detector gaps (Abadias et al., 2018). Thus for scalability, it is better suited to prepare TPV with emitter-detector separated by initial larger gaps and then electrostatically bring them closer.

Making an electrostatic actuation system, as the one shown in this work, contrary to reviewer’s concern, are inexpensive and trivial to mass-produce for existing foundries (Fedder et al., 2008; Fischer et al., 2015). The NEMS fabrication process is a mature technology developed over past several decades and widely implemented across variety of applications including sensing, micro-mirrors-based projection systems, inkjet printer heads and, RF switches.

Our efforts in this work tackle the long-standing challenge on scalability of TPV. Our demonstration is the first monolithically integrated TPV platform that uses a standard silicon fabrication process line, thereby making it practically scalable. The works demonstrated till date rely on a table top TPV system with an AFM-tip-emitter positioned at nano-scale distances from the detector using precision positioners (ref. 31) (Fiorino et al., 2018b, 2018a), which by no means are scalable. The only miniaturized TPV demonstrated uses wafer-bonding technique to individually bond a III-V detector and a silicon emitter (Inoue et al., 2019). In contrast, our platform relies on integrated NEMS that are, despite the reviewer’s claim, easily scalable. As

such it is our opinion that all these demonstrations (including ours) offer wildly different contributions to the field and can't be directly compared.

“... To ‘turn the system off’ one should (simply) remove the flow of heat to the TPV device (the only real option to avoid any additional inefficiency). ...”

A reconfigurable NEMS based TPV platform is indeed necessary as turning off heat source to a bulk system is not always feasible in practical applications requiring dynamic temperature management. For instance, heat recovery systems employed for cooling applications would require configurability to avoid unwanted and untimely cooling. In data centers as well as metal foundries, the local-temperature fluctuations are dynamically changing and therefore require reconfigurable heat-recovery (Boucher et al., 2006; Guha et al., 2012; DeSutter et al., 2019).

However, we agree, that re-configurability is not the central achievement here. As such we have removed the claim from the manuscript title and toned down this claim in the abstract.

“... However, the platform described by Bhatt et al. falls far short of providing a suitable and precise measured range of distances to critically evaluate device function. In fact, the distance data reported (Fig 3 and 4) lack the independent confirmation and accuracy of previous studies (see for example ref. 32). ...”

The distance measurement technique used here for estimating the emitter-detector gap is widely deployed for characterizing RF response of capacitive displacement sensors and switches (ref. 39-40 in main-text) (Fernández et al., 2006; Rottenberg et al., 2004; Zhang and Liao, 2012).

We have re-evaluated the distance measurement data based on the mentioned citations as discussed in the mentioned references improving our accuracy in estimation ($\sigma_{gap} \sim 30$ nm). The error bars in the gap-estimation are based on capacitance error that comes from a measurement performed over a frequency band (~ 4 -7 MHz).

The details of measurement, raw-data and assumptions for the fit performed are added in the supplementary section 7.

“... The reported platform is, even by the standards of this young field, inefficient ($\sim 0.3 \cdot 10^{-4}$) as compared to recently published work (for example ref. 32), and the assertion by that authors that the electrostatic control of the NEMS actuator consumes little power is not reassuring, because the actuation is not desirable or improving practical TPV platforms. ...”

It is important to point out that the work published in ref 32 (et al.) uses a commercially available III-V detector which is bonded to a silicon emitter at a fixed distance. In our case, the Ge photovoltaic detector, NEMS and the emitter are monolithically fabricated on single silicon substrate. In our case, the efficiency of our Ge photovoltaic is limited by the detector's low open-circuit voltage (V_{OC}), arising from high recombination rate in the Ge substrate. Considering the current state-of-art Ge PV quantum efficiency of over 10%, we can significantly improve ($> 4 \text{ mW} \cdot \text{cm}^{-2}$) the overall efficiency of our TPV (Korun and Navruz, 2016).

The NEMS actuator in our case consumes almost four orders of magnitude lower power than the power we generate.

“... With regard to the writing and presentation quality of the manuscript, it is difficult to fully comprehend the design of the NEMS system in relation to the photodetector and critically evaluated important aspects of the platform, including the thermal design and nanometer-level actuation of the system. ...”

The details of the structure, effect of actuation of NEMS on diode performance, temperature of the diode when emitter is hot, equivalent thermal circuit of the structure, details of actuation actuator performance are provided in the supplementary section.

“... Critical data, for example, on the cleanliness (particle contamination) and flatness of the devices are missing, and I am not at all convinced that the device is truly scalable to large areas. ...”

We have performed surface roughness measurements of the emitter and detector and found them to be $\sigma_{Det} \sim 1.1$ nm and $\sigma_{Emit} \sim 1.7$ nm, respectively. The confidence intervals shown in Fig 4b incorporates the contribution to heat transfer due to bending of the suspended structure under the influence of actuation potential. The calculations details are provided in supplementary section 2, the AFM measurement for roughness are provided in supplementary section 3.

“... Figure 1 is not clear, there should be a schematic drawing. ...”

In addition to the schematic shown in Fig. 1 a detailed schematic on the device is also provided in supplementary section 1.

Reviewer #3 (Remarks to the Author):

The topic of the manuscript, namely that of heat recycling using controlled near-field thermophotovoltaics, is very important and relevant, enabled by the recent advances in precision nanofabrication technologies. Indeed, the nanostructure complexity and precision in the control of the distance between the heater and the photodetector at the nanoscale, demonstrated in this manuscript, are impressive. The main results, reported in Fig 4b, show a super-linear increase in the generated power versus the decreasing gap distance. The measurement is carried out in 6 points only with rather large error bars both in the abscissa and ordinate axes. I apologize for my possible omission, but I couldn't see how the error bars were considered in the report of the measurement? In particular, why does the error bar grow with the increase in the signal amplitude (generated power)? Furthermore, two graphs based on the calculation (dotted red and blue lines) are reported to give the 'confidence interval' on the temperature of the hot side. However, I see that neither of the calculated curves fits the entire data set in a satisfactory manner (even including large error bars). Does this indicate systematic issues with the measurement of the generated power or the estimate of the hot temperature of the metal or both? I believe authors should offer some explanations for the large discrepancy between the measurement and the theory, as it relates directly to the main presented dataset.

The interpretation of the data strongly relies on the knowledge of the hot (T_{hot}) and cold (T_{cold}) reservoir temperatures. To the best of my knowledge, I did not see a reported measurement of the T_{hot} in the manuscript nor supplemental information (only an estimate based on the dissipated power into the W strip). The cold side is estimated in temperature from model fits of the I-V curves. I do not know what instrumentation is available to the authors, but given how important is the knowledge of these temperatures, I would like to see a more elaborated discussion of the measurement(s)/estimate(s)/fit(s) of these quantities, together with the confidence intervals (especially, in connection to the discussion of Fig 4b above). Given large error bars in the data, I believe that the claimed factor of 11 enhancement in the generated electrical power (in the near-field), should be specified with the experimental certainty bounds as well.

On a similar note, I found it quite astonishing that the temperature difference between a $\sim 900\text{K}$ and 300K body can be maintained so well when the bodies are in a perfect mechanical contact with each other over few-micron distances. As authors point out, this should only be possible by a "thin-film metallic emitter with high thermal resistance" and I think this is a discovery in itself that the authors found such a highly efficient thermal insulation. For clarity, I would like to understand about the details of the measurements. Are these steady-state measurements or obtained via modulation techniques (e.g. lock-in amplification)? Also, on what timescales are these measurements taken?

I would like to request mandatory revisions addressing the points raised. I believe the manuscript is otherwise very well written, clear and addresses an interesting topic, worthy of the attention of the broad readership of Nature Communications.

Response:

We thank the reviewer for their constructive feedback and appreciate the technical concerns raised. We have addressed the questions raised by the reviewer in a point-by-point basis below. These discussion made here are also added to the main-text at appropriate locations.

“... but I couldn't see how the error bars were considered in the report of the measurement? In particular, why does the error bar grow with the increase in the signal amplitude (generated power)? Does this indicate systematic issues with the measurement of the generated power or the estimate of the hot temperature of the metal or both? ...”

The error bars reported in generated power are computed based on multiple measurements of detector I_{sc} , while the error bars in the gap-estimation are based on capacitance error that comes from a measurement performed over a frequency band. The large deviation in measured power at smaller gaps (higher gate-potential) could be due to variation in electrical resistance under beam-stress leading to variations in delivered power (and temperature) and local variations in diode temperatures.

“... but given how important is the knowledge of these temperatures, I would like to see a more elaborated discussion of the measurement(s)/estimate(s)/fit(s) of these quantities, together with the confidence intervals (especially, in connection to the discussion of Fig 4b above). ...”

The temperature estimation of the emitter is performed based on independent measurement of temperature coefficient of resistance (TCR) and real-time estimation of the initial resistance R_0 using IV characteristics of emitter.

The TCR of pre-annealed ($t = 4$ hrs, $T = 500$ °C) emitters is measured on multiple TPV devices before suspending the bridge-emitters using a calibrated temperature controlled chuck mount. The measurement of resistance is performed at different temperatures inside the vacuum probe station. Using the slope of the measured T vs R plot, the TCR (α_E) is estimated.

$$\frac{R_T - R_0}{T - T_0} = \alpha_E \cdot R_0$$

Where R_T is the measured resistance at temperature T and R_0 and T_0 , are the initial resistance and temperature, respectively.

Estimation of R_0 is performed real-time during the heating of the emitter. Un-annealed and suspended emitter chips are gradually heated up by applying electrical current. The electrical power delivered to the emitters is computed using the simultaneous measurement of voltage. The temperature ramp-up is performed relatively faster initially but gradually reduced it to a rate of ~ 100 °C per hour. The slower rate is to avoid thermal shock to the suspended structures and allow enough time for the resistance to stabilize through the process of self-anneal. The entire heating up process takes approximately 4 hours. We sweep the input power in a small range around the parked temperature/power and simultaneously record the resistance. Using P vs R plot, we estimate the R_0 and R of the emitter.

The details of this measurement along with measured data and, fit parameters are provided in supplementary section 7.

““... Given large error bars in the data, I believe that the claimed factor of 11 enhancement in the generated electrical power (in the near-field), should be specified with the experimental certainty bounds as well. ...””

We have modified our claim on the enhancement factor by adding an experimental certainty bounds.

““... For clarity, I would like to understand about the details of the measurements. Are these steady-state measurements or obtained via modulation techniques (e.g. lock-in amplification)? Also, on what timescales are these measurements taken? ...””

The results reported here are obtained through steady state measurements performed using dedicated source meters for the heaters, the detector and the NEMS gate. Once the desired temperature is achieved (i.e. temperature at which detector current is high enough to continue the NEMS measurements), the NEMS gate is actuated. The diode IV characteristics are recorded at different actuation potentials (V_a) with an entire measurement taking about 15 minutes. The source meter used for measuring the diode IV characteristics is set to an integration time of ~ 80 ms/sample and an averaging filter set at 25 samples/data-point.

The measurement setup and circuit connections are added to the methods section.

References.

1. Abadias, G., Chason, E., Keckes, J., Sebastiani, M., Thompson, G.B., Barthel, E., Doll, G.L., Murray, C.E., Stoessel, C.H., Martinu, L., 2018. Review Article: Stress in thin films and coatings: Current status, challenges, and prospects. *J. Vac. Sci. Technol. A* 36, 020801. <https://doi.org/10.1116/1.5011790>
2. Boucher, T.D., Auslander, D.M., Bash, C.E., Federspiel, C.C., Patel, C.D., 2006. Viability of Dynamic Cooling Control in a Data Center Environment. *J. Electron. Packag.* 128, 137–144. <https://doi.org/10.1115/1.2165214>
3. DeSutter, J., Tang, L., Francoeur, M., 2019. A near-field radiative heat transfer device. *Nat. Nanotechnol.* 14, 751–755. <https://doi.org/10.1038/s41565-019-0483-1>
4. Fedder, G.K., Howe, R.T., Liu, T.-J.K., Quevy, E.P., 2008. Technologies for Cofabricating MEMS and Electronics. *Proc. IEEE* 96, 306–322. <https://doi.org/10.1109/JPROC.2007.911064>
5. Fernández, L.J., Wiegerink, R.J., Flokstra, J., Sesé, J., Jansen, H.V., Elwenspoek, M., 2006. A capacitive RF power sensor based on MEMS technology. *J. Micromechanics Microengineering* 16, 1099–1107. <https://doi.org/10.1088/0960-1317/16/7/001>
6. Fiorino, A., Thompson, D., Zhu, L., Song, B., Reddy, P., Meyhofer, E., 2018a. Giant Enhancement in Radiative Heat Transfer in Sub-30 nm Gaps of Plane Parallel Surfaces. *Nano Lett.* 18, 3711–3715. <https://doi.org/10.1021/acs.nanolett.8b00846>
7. Fiorino, A., Zhu, L., Thompson, D., Mittapally, R., Reddy, P., Meyhofer, E., 2018b. Nanogap near-field thermophotovoltaics. *Nat. Nanotechnol.* 13, 806–811. <https://doi.org/10.1038/s41565-018-0172-5>
8. Fischer, A.C., Forsberg, F., Lapisa, M., Bleiker, S.J., Stemme, G., Roxhed, N., Niklaus, F., 2015. Integrating MEMS and ICs. *Microsyst. Nanoeng.* 1, 1–16. <https://doi.org/10.1038/micronano.2015.5>
9. Guha, B., Otey, C., Poitras, C.B., Fan, S., Lipson, M., 2012. Near-Field Radiative Cooling of Nanostructures. *Nano Lett.* 12, 4546–4550. <https://doi.org/10.1021/nl301708e>
10. Inoue, T., Koyama, T., Kang, D.D., Ikeda, K., Asano, T., Noda, S., 2019. One-Chip Near-Field Thermophotovoltaic Device Integrating a Thin-Film Thermal Emitter and Photovoltaic Cell. *Nano Lett.* 19, 3948–3952. <https://doi.org/10.1021/acs.nanolett.9b01234>
11. Kloppstech, K., Könné, N., Biehs, S.-A., Rodriguez, A.W., Worbes, L., Hellmann, D., Kittel, A., 2017. Giant heat transfer in the crossover regime between conduction and radiation. *Nat. Commun.* 8, 1–5. <https://doi.org/10.1038/ncomms14475>
12. Korun, M., Navruz, T.S., 2016. Comparison of Ge, InGaAs p-n junction solar cell. *J. Phys. Conf. Ser.* 707, 012035. <https://doi.org/10.1088/1742-6596/707/1/012035>
13. Markel, V.A., 2016. Introduction to the Maxwell Garnett approximation: tutorial. *JOSA A* 33, 1244–1256. <https://doi.org/10.1364/JOSAA.33.001244>
14. Reboud, V., Gassenq, A., Hartmann, J.M., Widiez, J., Virost, L., Aubin, J., Guillois, K., Tardif, S., Fédéli, J.M., Pauc, N., Chelnokov, A., Calvo, V., 2017. Germanium based photonic components toward a full silicon/germanium photonic platform. *Prog. Cryst. Growth Charact. Mater.* 63, 1–24. <https://doi.org/10.1016/j.pcrysgrow.2017.04.004>
15. Rottenberg, X., Brebels, S., Raedt, W.D., Nauwelaers, B., Tilmans, H.A.C., 2004. RF-power: driver for electrostatic RF-MEMS devices. *J. Micromechanics Microengineering* 14, S43–S48. <https://doi.org/10.1088/0960-1317/14/9/007>

16. Serry, F.M., Walliser, D., Maclay, G.J., 1998. The role of the casimir effect in the static deflection and stiction of membrane strips in microelectromechanical systems (MEMS). *J. Appl. Phys.* 84, 2501–2506. <https://doi.org/10.1063/1.368410>
17. Xu, D.Y., Bilal, A., Zhao, J.M., Liu, L.H., Zhang, Z.M., 2019. Near-field radiative heat transfer between rough surfaces modeled using effective media with gradient distribution of dielectric function. *Int. J. Heat Mass Transf.* 142, 118432. <https://doi.org/10.1016/j.ijheatmasstransfer.2019.118432>
18. Zhang, Z., Liao, X., 2012. GaAs MMIC fabrication for the RF MEMS power sensor with both detection and non-detection states. *Sens. Actuators Phys., Selected papers from The 16th International Conference on Solid-State Sensors, Actuators and Microsystems* 188, 29–34. <https://doi.org/10.1016/j.sna.2012.01.020>

REVIEWERS' COMMENTS:

Reviewer #1 (Remarks to the Author):

The questions and concerns I raised in my original report have been carefully addressed by the authors. I do not have any further questions.

I would like to point out that I respectfully disagree with the assessment made by Reviewer 2. I believe that the reconfigurable platform (tunable gap) is worthy of investigation, and certainly worthy of publication. I think that it is premature to claim at this point that a reconfigurable near-field TPV platform is "misguided" and "wrong". From a technical standpoint, the current manuscript is a significant contribution to near-field TPV device development. I think that the scientific community should judge whether or not an adjustable platform is the future of near-field TPV.

In summary, I support publication of this manuscript in Nature Communications. I believe that this work is a significant contribution to the field of near-field TPV energy conversion.

Reviewer #3 (Remarks to the Author):

I am satisfied with the explanations given by the authors and recommend the manuscript for publication.